# Camrelizumab-Induced Isolate Abducens Neuritis: A Rare Ophthalmic Immune-Related Adverse Events

**DOI:** 10.3390/brainsci12091242

**Published:** 2022-09-14

**Authors:** Yanli Hou, Qiang Su, Simeng Tang, Hongyang Li

**Affiliations:** 1Department of Ophthalmology, Beijing Friendship Hospital, Capital Medical University, Beijing 100050, China; 2Department of Oncology, Beijing Friendship Hospital, Capital Medical University, Beijing 100050, China

**Keywords:** immunotherapy, abducens neuritis, Camrelizumab, immune-related adverse events, case report

## Abstract

Background: Anti-tumor immunotherapy with immune checkpoint inhibitors induces several immune-related adverse events. Camrelizumab-related isolate abducens neuritis is rare. Case presentation: We report on a 67-year-old man with esophageal cancer who presented with acute-onset isolated right abducens cranial nerve palsy after ten cycles of Camrelizumab treatment. Magnetic resonance imaging examination revealed thickening and post-contrast enhancement at the cisternal segment of the right abducens nerve. The diagnosis was immune-related abducens neuritis caused by Camrelizumab. We put him on oral taper corticoids (methylprednisone) for neuritis treatment without Camrelizumab suspension. One month after treatment, he recovered completely. At the last follow-up, one year after the onset of diplopia, the patient was in good condition without neurological symptom recurrence. Conclusion: Abducens neuritis is a rare immune-related adverse outcome of Camrelizumab. The present case proves the efficacy and safety of using corticoids in the treatment of abducens neuritis.

## 1. Introduction

Abducens nerve palsy, caused by various conditions, is a common neuro-ophthalmology outcome. Most cases are approximately 70 years old, and microvascular ischemia is the leading etiology [1]. Neuritis of the isolated abducens nerve is relatively rare. According to an early 1984 report [2], the incidence of abducens nerve palsy was 11.3/100,000 per year in the US, and only 8% of these cases were caused by inflammation.

Immune checkpoint inhibitors (ICIs) are novel immunologic agents that have revolutionized oncological practice [3]. ICIs alleviate tumor-induced T-cell inhibition, potentiate T-cell activation, and evoke anti-tumor immune attacks. Camrelizumab is a programmed death receptor 1 (PD-1) inhibitor. It received its first global approval in 2019 in China [4]. Several phase III trials [5] have demonstrated that, compared with chemotherapy, ICI treatment facilitates a significantly longer overall survival in previously treated patients with advanced esophageal cancer. Unfortunately, ICIs induced several immune-related adverse events (irAEs) due to their immune-enhanced nature. Camrelizumab has induced several immune-related adverse events (irAEs) such as pneumonitis [6], hepatitis [7], myocarditis [8] and myositis [9] in recent reports. Ophthalmic irAEs commonly manifest as uveitis (1%) and dry eye (1–24%) [10]; 83.6% of cases of ICI-related uveitis were diagnosed within six months [11].

In this study, we describe a patient with esophageal cancer who manifested isolated abducens neuritis during anti-tumor immunotherapy. Magnetic resonance imaging (MRI) can localize and characterize the lesions and is an excellent modality diagnostic tool for the abducens nerve disorder. Abducens neuritis may be an irAE of Camrelizumab.

## 2. Case Presentation

A 67-year-old man with esophageal cancer presented with acute-onset horizontal diplopia for two days after ten cycles of Camrelizumab treatment. The patient had no complications of ocular pain or headaches. His ocular examination was normal. A neurological examination showed isolated right abducens cranial nerve palsy. The right eye could not abduct completely, which was confirmed by the Hess screen (Figure 1). An MRI examination revealed thickening and post-contrast enhancement at the cisternal segment of the right abducens nerve (Figure 2). There was no evidence of brain tumor metastases, cerebrovascular disorder, or orbital lesions. The laboratory tests showed negative antibodies to human immunodeficiency virus (HIV)/Hepatitis C virus (HCV), rapid plasma regain (RPR), and Hepatitis B surface antigens (HBsAg).

The patient was diagnosed with middle-lower section esophageal cancer with lymph node metastasis one year ago (Figure 3) (esophageal poorly differentiated adenocarcinoma; immunohistochemistry result: CK5/6(−); CK7(+); P40(−); Ki-67(>50%+); CDX-2(−); MSH2(+); MSH6(+); MLH1(+); TTF-1(+); NapsinA(+); P63(−)) (Figure 4).

The patient had no history of hypertension, diabetes, heart disease, hyperlipidemia, cerebrovascular disease or autoimmune disease. He accepted radiotherapy and chemotherapy (Albumin-bound paclitaxel combined with carboplatin/Cisplatin/Nedaplatin). The chemotherapy drugs had to be changed frequently due to severe tertiary hemopenia, and the cancer was not well controlled until combined immunotherapy (Camrelizumab) was performed six months ago (Figure 5). After this, he was in good physical condition but had diplopia. The levels of the serum tumor marker were normal (CEA: 2.18 ng/mL; CYFRA211: 2.8 ng/mL; NSE: 12.33 ng/mL; Pro-Grp: 32.71 pg/mL). It was interpreted as immune-related abducens neuritis caused by Camrelizumab. The paraneoplastic syndrome-associated autoantibodies in the serum were negative. (anti-Hu, Ri, CV2, Amphiphysin, Ma1, Ma2, SOX1, Tr, Zic4, GAD65, PKCγ, Recoverin, Titin).

We put the patient on oral corticoids (methylprednisone, 48 mg/d) for neuritis treatment after the diagnosis, and a subsequent taper was also administered. Prior to methylprednisone therapy, the patient was treated with enhanced microvascular perfusion drugs, but these failed to work. Camrelizumab was continued without interruption. After one week of treatment, the ability of the right eye to abduct partly recovered. One month later, the diplopia was completely resolved. (Figure 1). At the last follow-up, one year after the onset of diplopia, the patient was in good condition without tumor progression or neurological symptom recurrence.

Importantly, potentially toxic drugs were adopted before ICIs in this case. However, paclitaxel combined with platinum was administered almost fourteen cycles before the onset of abducens neuritis, which appeared abruptly and was resolved with corticosteroids, suggesting that an inflammatory rather than toxic etiology complicated ICI treatment.

Only a few case reports have reported abducens neuritis following PD-1 as possible complications (Table 1), and therefore, information on the spectrum, treatment and outcome of PD-1-induced abducens neuritis is limited.

## 3. Discussion

Cranial nerve palsy caused by neuritis is characterized by thickening and post-contrast enhancement, shown under high-resolution MRI [15]. The abducens nerve is the sixth cranial nerve. It is more frequently associated with III, IV, V and VII nerve pathology. The etiologies of acquired abducens nerve palsy include microvascular ischemia, vasculopathy, central nervous system neoplasm, multiple sclerosis, trauma, stroke, and infection [16]. However, 13–35% of cases are idiopathic [17]. Conversely, anti-tumor immunotherapy with ICIs provided inflammatory conditions that could be potential clinical courses for neuritis. In this study, we present a case of isolated abducens neuritis induced by Camrelizumab, diagnosed according to diplopia, failure of abducting, and characterized MRI images. As we know, neuritis and ischemic neuropathy are difficult to distinguish in MRI, and we cannot completely exclude the possibility of ischemia. However, the nerve thickening and post-contrast enhancement on MRI cannot be seen in ischemic lesions but in neuritis. This patient has no vasculopathic risk factors or cerebrovascular disorders, so the drug is the most likely cause. 

The ICI-induced central and peripheral nerve system irAEs were found in approximately 1–3% of the patients [18]. The possible complications included limbic encephalitis, meningoencephalitis, cerebellitis, peripheral neuropathy, myasthenia gravis, and myopathy [19,20]. The ICI-induced cranial nerve irAEs were rarely described. The irAEs occur between 1 and 6 months after the initiation of the ICI. Sun et al. demonstrated that cranial nerve irAEs are less common than optic nerve but occur earlier. Vogrig et al. reported that two-thirds of cranial neuritis could be improved or resolved by corticoids, and irAEs might well be responsible for the tumor cells’ response to immunotherapy [12]. Conversely, Sun et al. [14] found that cranial neuritis responses to steroid treatment were variable. They have more propensity for poorer ocular prognosis with a higher mortality rate. Consistent with previous cases, abducens neuritis occurs commonly in skin malignant melanoma during anti-PD-1 therapy. This is because melanoma was the first and the most common indication for treatment with ICIs. Moreover, little is known regarding ICI-induced isolated cranial nerve palsy. Information on the spectrum, treatment and outcome are unavailable. 

It is challenging to balance antitumor-immunotherapy and irAE treatment. Fortunately, this abducens neuritis patient recovered quickly and showed a favorable cancer prognosis. The corticoid treatment lasted for a short duration and without antitumor immunotherapy suspension. Larger-scale and longer-term studies are needed to investigate and confirm the natural course of ICI-induced isolated abducens neuritis.

## 4. Conclusions

Abducens neuritis is a rare irAE of Camrelizumab. The present case proves the efficacy and safety of using corticoids in the treatment of abducens neuritis.

## Figures and Tables

**Figure 1 brainsci-12-01242-f001:**
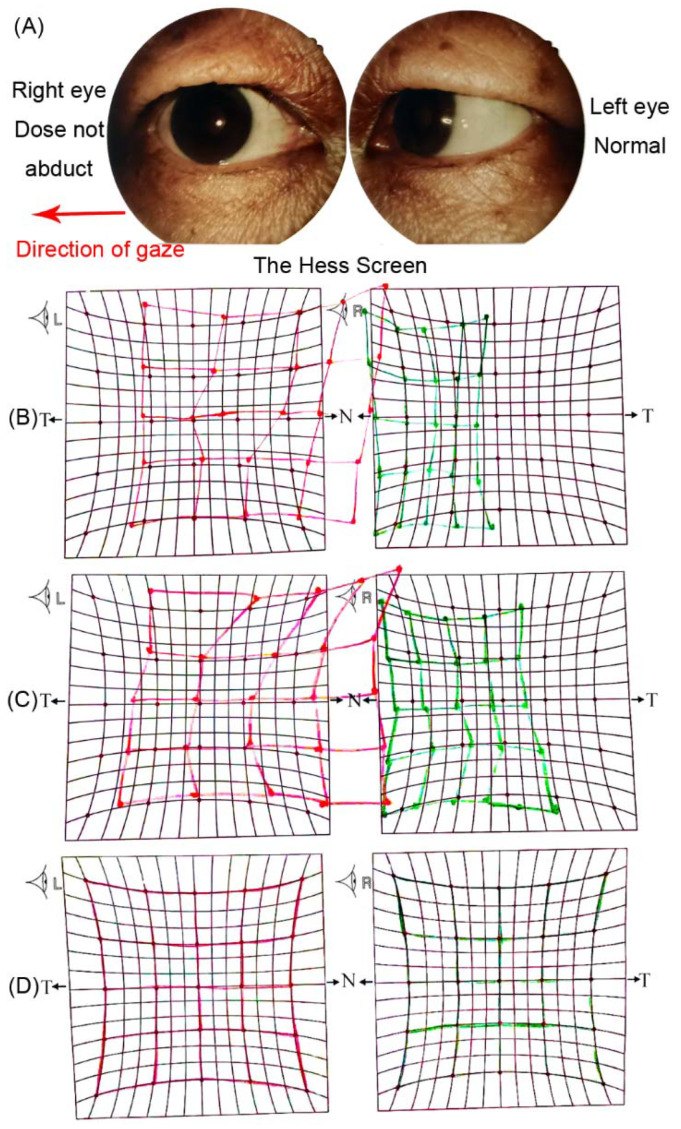
(**A**) The right eye does not abduct when gazing in the right direction. The HESS screen test on the onset days (**B**), one week (**C**), and one month (**D**) after corticoid treatment. T: temporal, N: Nasal.

**Figure 2 brainsci-12-01242-f002:**
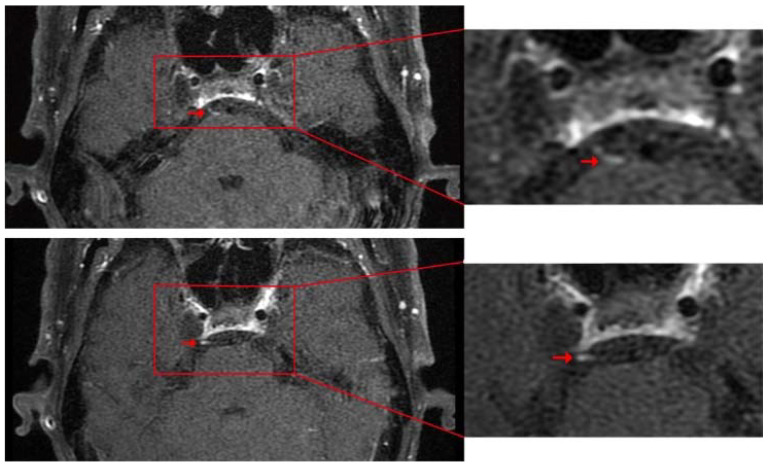
Post-contrast Axial T1WI. There is marked enhancement and thickening of the right abducens nerve (right arrow).

**Figure 3 brainsci-12-01242-f003:**
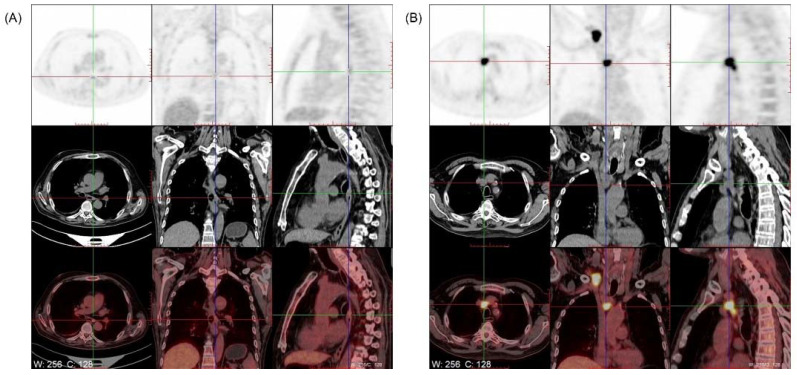
18F-FDG-PET of the patient. (**A**) Esophageal cancer (after treatment): High accumulation in the middle-lower esophageal section with a maximum SUV of 3.8. (**B**) Lymph node metastasis: High accumulation in the mediastinum lymph node (region 2R 1.8 cm × 2.2 cm) with a maximum SUV of 17.8.

**Figure 4 brainsci-12-01242-f004:**
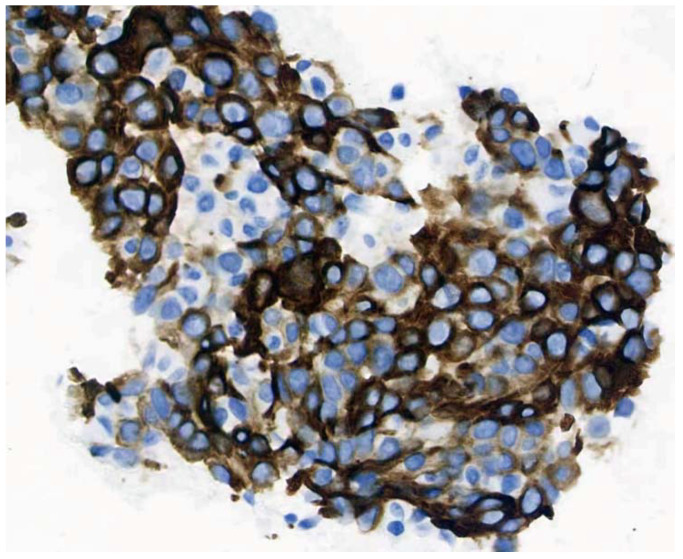
Immunohistochemistry result of tumor cells (40 × 10): CK7(+).

**Figure 5 brainsci-12-01242-f005:**
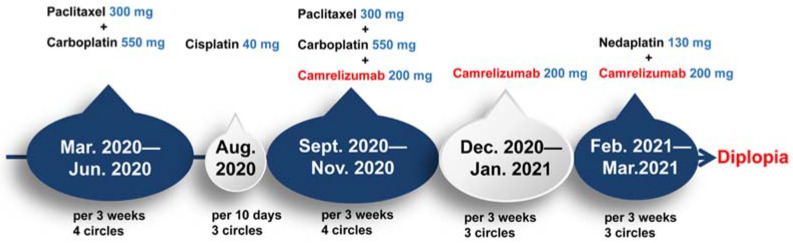
A detailed clinical timeline is displayed. In March 2020, the patient was hospitalized. After a series of examinations, the patient was diagnosed with middle-lower section esophageal cancer with lymph node metastasis. In March 2020, the patient was started on a chemotherapy regimen containing paclitaxel/carboplatin for four cycles. The regimen was subsequently switched to Cisplatin for three cycles. From September 2020, Camrelizumab was arranged in ten cycles with or without the combination of chemotherapy. The horizontal diplopia was onset in April 2021.

**Table 1 brainsci-12-01242-t001:** Cases of abducens neuritis reported with anti-PD-1 immunotherapy.

Author/Year	Gender	Age	Tumour	Anti-PD-1	Treatment of irAE	Outcome of irAE	Other irAE
Vogrig, [12] 2021	Male	63	Melanoma	Ipilimumab- pembrolizumab	pause of ICIs oral corticosteroids	Resolved	autoimmune colitis
	Male	82	Melanoma	Nivolumab	pause of nivolumab prednisone (1 mg/kg/day)	Resolved	facial nerve palsy
Zimmer, [13] 2016	Male	83	Melanoma	Nivolumab	pause of nivolumab Methylprednisolone (1 mg/kg/day)	Resolved	facial nerve palsy
Sun, [14] 2021	Male	52	Melanoma	Ipilimumab/ nivolumab	Methylprednisolone; prednisone	Improved	NA

irAE: immune-related adverse events, ICIs: immune checkpoint inhibitors.NA: not available.

## Data Availability

All of the data in this study are in the article.

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
