# Peer review of "Camrelizumab-Induced Isolate Abducens Neuritis: A Rare Ophthalmic Immune-Related Adverse Events"

_brainsci, 2022, doi:10.3390/brainsci12091242_

Round 1

Reviewer 1 Report

Extensive editing of English language and style required. Otherwise the paper is difficult to read

It is not clear how authors excluded other causes such as ischemia or paraneoplastic syndrome.

Author Response

Point 1: Extensive editing of English language and style required. Otherwise the paper is difficult to read

Response 1: Thank you for the suggestion. We submitted the manuscript for English editing at the MOPI English editing system, and with their help, the revised manuscript is easier to understand now.

Point 2: It is not clear how authors excluded other causes such as ischemia or paraneoplastic syndrome.

Response 2: Thank you for the insightful comment. It is crucial to discriminate cranial nerve neuritis from ischemic changes of the cranial nerve. We excluded ischemia or paraneoplastic syndrome for the following reason. First, as we see in Figure 2, a high-resolution MRI examination revealed thickening and post-contrast enhancement at the cisternal segment of the patient’s right abducens nerve, it cannot be seen in ischemic lesions. Second, there was no evidence of cerebrovascular disorders on MRI and the patient has no vasculopathic risk factors. Third, before methylprednisone therapy, improvement of microvascular perfusion doesn’t work. Forth, the paraneoplastic syndrome-associated autoantibodies in the serum were negative. (anti-Hu, Ri, CV2, Amphiphysin, Ma1, Ma2, SOX1, Tr, Zic4, GAD65, PKCγ, Recoverin, Titin)

We add the new information in the Case Presentation part of the revised manuscript.

Reviewer 2 Report

Dear Authors, 

Great work with the case report. I do agree reporting adverse events is imperative and this case report also presents the management of the patient post diagnosis. I would however, recommend authors to provide the immunohistochemsitry images (hig resolution) as documented in line 74-76. In addition, I would also want authors to provide a tabular data of any cases of abducens neuritis reported with anti-PD1 immunotherapy. I would hope authors would appreciate the fact that its not the immunotherapy agent e.g. Camrelizumab as in this article, but the mechanism of action of immune checkpoint inhibitor e.g. Camrelizumab being anti-PD1 inhibitor. I would want to see collation of data from other anti-PD1 immunotherapies in the field and provide a comparative incidence of anducens neutritis with check-point therapies. 

Author Response

Point 1: Great work with the case report. I do agree reporting adverse events is imperative and this case report also presents the management of the patient post diagnosis. I would however, recommend authors to provide the immunohistochemsitry images (hig resolution) as documented in line 74-76. In addition, I would also want authors to provide a tabular data of any cases of abducens neuritis reported with anti-PD1 immunotherapy. I would hope authors would appreciate the fact that its not the immunotherapy agent e.g. Camrelizumab as in this article, but the mechanism of action of immune checkpoint inhibitor e.g. Camrelizumab being anti-PD1 inhibitor. I would want to see collation of data from other anti-PD1 immunotherapies in the field and provide a comparative incidence of anducens neutritis with check-point therapies. 

Response 1: Thank you for the insightful comment. First, the patient does have the immunohistochemistry high-resolution images of the tumour cells but we need one more week to download them from Pathology. We will add the images documented in lines 74-76 as soon as we get them. 

       Second, only a few case reports have reported abducens neuritis following PD-1 as possible complications (Table 1), and therefore, information on the spectrum, treatment and outcome of PD-1-induced abducens neuritis is limited.

We add Table 1 in the Case Presentation part of the revised manuscript.

Table1 Cases of abducens neuritis reported with anti-PD-1 immunotherapy
Author/Year Gender  Age Tumour Anti-PD-1 antibody Treatment of  irAE Outcome of  irAE other  irAE
Vogrig, 2021 Male 63 Melanoma Ipilimumab-pembrolizumab pause of ICIs Resolved autoimmune colitis
          oral  corticosteroids    
  Male 82 Melanoma Nivolumab pause of nivolumab Resolved facial nerve palsy
          prednisone (1 mg/kg/day)    
Zimmer, 2016 Male 83 Melanoma Nivolumab  pause of nivolumab Resolved facial nerve palsy
          Methylprednisolone 1 mg/kg/d    
Sun, 2021 Male 52 Melanoma Ipilimumab/nivolumab Methylprednisolone; prednisone Improved NA
 irAE: immune-related adverse events, ICIs:immune checkpoint inhibitors 

Reviewer 3 Report

Interesting observation. The English needs to be reviewed for grammar, tenses and ambiguity.

For the discussion it is important to discuss the concept of "neuritis" , which is not well supported. No CSF, no immune markers. It could well be vascular.

The discussion is too long an repeats previous content, and also goes too far into other details.

In summary  worthwhile to report, major revision needed.

Author Response

Dear Editor,

      We have studied the valuable comments from you and the reviewers carefully. We tried our best to revise the manuscript. We submitted the manuscript for English editing at the MOPI English editing system, and with their help, the revised manuscript is easier to understand now. The point-to-point responses to the reviewer’s comments are listed as follows:

We also have carefully corrected all mistakes and new table/figure were added.

Point 1: Interesting observation. The English needs to be reviewed for grammar, tenses and ambiguity.

Response 1: Thank you for the suggestion. We submitted the manuscript for English editing at the MOPI English editing system, and with their help, the revised manuscript is easier to understand now.

Point 2: For the discussion it is important to discuss the concept of "neuritis" , which is not well supported. No CSF, no immune markers. It could well be vascular.

Response 2: Thank you for the insightful comment. It is crucial to discriminate cranial nerve neuritis from ischemic changes of the cranial nerve. We excluded ischemia or paraneoplastic syndrome for the following reason. First, as we see in Figure 2, a high-resolution MRI examination revealed thickening and post-contrast enhancement at the cisternal segment of the patient’s right abducens nerve, it cannot be seen in ischemic lesions. Second, there was no evidence of cerebrovascular disorders on MRI and the patient has no vasculopathic risk factors. Third, before methylprednisone therapy, improvement of microvascular perfusion doesn’t work. Forth, the paraneoplastic syndrome-associated autoantibodies in the serum were negative. (anti-Hu, Ri, CV2, Amphiphysin, Ma1, Ma2, SOX1, Tr, Zic4, GAD65, PKCγ, Recoverin, Titin). Fifth, the CSF examination was not performed because the patient recovered quickly after corticoid therapy, and the CSF examination was invasive.

We add the new information in the Case Presentation part of the revised manuscript.

Point 3:The discussion is too long an repeats previous content, and also goes too far into other details.

Response 3: Thank you for the insightful comment. We have improved the Discussion as you suggested in the revised manuscript. The duplicate sections were removed and some new literature and case reports were added.

Round 2

Reviewer 1 Report

no other comments

Author Response

Dear Editor,

Thank you for your hard work. We have studied the valuable comments from you and the reviewers carefully. We tried our best to revise the manuscript. We added immunohistochemistry high-resolution images of the tumor cells. We have revised the discussion section. We also have carefully corrected English language and style mistakes.

Point 1: no other comments

Reviewer 2 Report

Accept

Author Response

Dear Editor,

Thank you for your hard work. We have studied the valuable comments from you and the reviewers carefully. We tried our best to revise the manuscript. We added immunohistochemistry high-resolution images of the tumor cells. We have revised the discussion section. We also have carefully corrected English language and style mistakes.

Point 1: Accept
